# Angiotensin II and Angiotensin Receptors 1 and 2—Multifunctional System in Cells Biology, What Do We Know?

**DOI:** 10.3390/cells10020381

**Published:** 2021-02-12

**Authors:** Maksymilian Ziaja, Kinga Anna Urbanek, Karolina Kowalska, Agnieszka Wanda Piastowska-Ciesielska

**Affiliations:** Department of Cell Culture and Genomic Analysis, Medical University of Lodz, Zeligowskiego 7/9, 90-752 Lodz, Poland; maksymilian.ziaja@stud.umed.lodz.pl (M.Z.); kinga.urbanek@umed.lodz.pl (K.A.U.); karolina.kowalska1@umed.lodz.pl (K.K.)

**Keywords:** angiotensin, renin-angiotensin system, cancer, pathophysiology

## Abstract

For years, the renin-angiotensin system (RAS) has been perceived as a system whose role is to primarily modulate the functioning of the cardiovascular system. Years of research into the role of RAS have provided the necessary data to confirm that the role of RAS is very complex and not limited to the cardiovascular system. The presence of individual elements of the renin-angiotensin (RA) system allows to control many processes, ranging from the memorization to pro-cancer processes. Maintaining the proportions between the individual axes of the RA system allows for achieving a balance, often called homeostasis. Thus, any disturbance in the expression or activity of individual RAS elements leads to pathophysiological processes.

## 1. Introduction

Traditionally, the renin-angiotensin system (RAS) has been viewed exclusively as a system involved in the regulation of salt and fluid homeostasis and blood pressure. Chronic RAS activation in the cardiovascular system increases angiotensin II (Ang II) ratio and plays a major role in the pathogenesis of cardiovascular diseases such as hypertension and heart failure. The RAS consists of the Ang II precursor angiotensinogen; renin; angiotensin-converting enzyme (ACE); Ang II; and Ang II type 1 and type 2 receptors (AT1R and AT2R). Ang II is the main effector molecule of the RAS, deriving from angiotensinogen by successive enzymatic actions of renin and ACE. Mounting evidence demonstrated that tightly controlled RAS activity is not only critical to maintaining systemic hemodynamics and blood volume but also for the control of cell proliferation, differentiation, invasiveness, metastasis, and tissue remodeling in target organs [1,2]. Such a wide range of influence carries a certain risk with it. Literature data unequivocally show a significant impact of mutations and single nucleotide polymorphism (SNP) of the ATR1 gene on upregulation of its product—AT1R, hence the disruption of the mentioned above cellular actions [3]. The balance between the activation of individual receptors allows to maintain the correct states between vasoconstriction and vasodilation, or between pro- and anti-inflammatory responses. Furthermore, functional, local RAS has been identified in many organs and tissues, including brain and pituitary gland [4,5,6], adrenal gland [7], prostate gland, endometrium, pancreas, mammary gland, ovaries, testis, and the epididymis [8,9,10,11,12]. In vitro studies also indicate that RAS is involved in mental health regulation through ATR1 activity modulation by proper antagonist. Additionally, the newly published data clearly show the potential role of the RAS in a wide range of pathologies resulting from disturbances in pancreatic endocrine homeostasis to endometrial cancer.

The aim of this review is to evaluate the multifunctional potential of angiotensin and angiotensin receptors in physiological state of various organs and directly in cancer development.

### 1.1. Angiotensin—A Multi Task Force Peptide Hormone

Currently, it is known that Ang II is not the only one active component of the RAS. The discovery of other biological active peptides such as angiotensin III (Ang III), IV (Ang IV), 1-7 (Ang 1-7), and enzymes leading to their formation, as well as new specific receptors AT1-7/Mas or AT4/ insulin-regulated membrane aminopeptidase (IRAP) shed a new light of the role of angiotensins. With these new findings, it became obvious that angiotensin family peptides can act not only endocrine, but also autocrine or paracrine. RAS, especially Ang II/AT1R and Ang IV/Mas signaling, is not just a regulator of cardio-renal physiology.

Undoubtedly, Ang II is the most important effector peptide for both systemic as well as local RAS. There are two pathways of forming of Ang II. The first one depends on angiotensin converting enzyme (ACE), which cleaves off the His-Leu dipeptide from the C-terminus of angiotensin I (Ang I), converts it into a key peptide of the RAS. Alternative metabolic pathways allow the generation of Ang II from Ang I or directly from angiotensinogen with the involvement of other proteolytic enzymes such as tonin, elastase, cathepsin C, G, E, or chymostatin-sensitive angiotensin II-generating enzyme (CAGE). As mentioned previously, Ang II can be converted to other biological active peptides by cutting off consecutive amino acids from the C of its terminus polypeptide chain. The functioning of the local RAS and its impacts on the tissues and organs largely depends on the balance between synthesis and metabolism of angiotensin peptides as well as the receptor levels sensitive to their action.

Ang II acts through two classic angiotensin receptors (ATRs), angiotensin receptor type 1 (AT1R) and angiotensin receptor type 2 (AT2R). ATRs belong to G-protein-coupled receptors that contain seven transmembrane domains and have molecular mass of 41 kDa. Despite the similar chain length of the receptors, the sequence homology between AT1R and AT2R is only 32–34%. AT1R consists of 359 amino acids and is located in brain, adrenal gland, kidneys, vasculature, and heart, while AT2R has 363 amino acids and is expressed during fetal development, in brain, adrenal glands, and reproductive tissues [13]. The effects of AT1R and AT2R stimulation are essentially contradictory. This is not only due to the fact of activation of different signaling pathways, but also from the heterodimerization of AT1R/AT2R, which alters the conformation and blocks AT1R. Maintaining the appropriate proportions, in the context of activation of individual receptor subtypes, determines the proper functioning of cells. Consequently, any changes leading to a disturbance of proportions lead to pathological conditions [14].

The existence of other receptor types, which are neither AT1R nor AT2R, has been proven. The pharmacological difference from the classic angiotensin receptor AT4R, has been defined as IRAP, also known as leucyl/cystinyl aminopeptidase or oxytocinase. The gene of the human AT4R/IRAP is located on chromosome 5q21. The receptor protein consists of 1025 amino acids with a calculated molecular mass of 165 kDa and is found in tissues like adrenal gland, heart and kidney, bladder, colon, lung, prostate, and vascular smooth muscle [15,16,17,18,19]. It has been observed that AT4R has a high-affinity binding site for the hexapeptide Ang IV but much smaller in relation to Ang III or Ang II. Furthermore, Ang IV-activated AT4R, similar to Ang II and AT2R, can present an antagonistic effect towards the classic AT1R.

As functional receptor for Ang 1-7, Mas protooncogene was mapped to the chromosomal region 6q24-27. This gene encodes a protein of 325 amino acids belonging to the largest class of membrane receptors—G protein-coupled receptors (GPCRs). The classic angiotensin receptors also include the 7-transmembrane receptors but have only 8% and 19% sequential homology with Mas protein, respectively. The AT1-7/Mas receptor is expressed in brain, testis, ovary, colon endothelial cells of blood vessels, and lungs [12,20,21,22]. Besides AT2R, the Mas receptor axis is also known to antagonize the effect of the AT1R axis. Moreover, literature data report that Mas can interact with both classic angiotensin receptors through heterodimerization [23].

AT1R and AT2R are supposed to play different roles during cancerogenesis. Ang II acting via AT1R is correlated with proliferative, anti-apoptotic, pro-angiogenic, pro-inflammatory, and metastatic actions and blockage of cell cycle in G1 phase, while AT2R is related to the attenuation of tumor growth, induction of apoptosis, and differentiation [24,25]. Moreover, ATR2 presents overall protective and regenerating actions like anti-inflammatory and anti-fibrotic properties [26]. 

Alternatively, many other Ang II-generating enzymes for which Ang I may act as a substrate have been identified. Nevertheless, it was stated that numerous Ang II metabolites, resulting from the action of proteolytic enzymes, have been defined as biologically active.

### 1.2. RAS in Health and Disease

Beyond the classical role of RAS, recently it has been noticed that this system correlates with cancerogenesis, modulation of glucose-stimulated insulin secretion [27], metabolic syndrome [28], progression of diabetic retinopathy, age-related macular degeneration [29], diabetes comorbid depression [30], and even migraine [31]. RAS influences processes like proliferation, apoptosis, autophagy, migration, inflammation, oxidative stress, or angiogenesis [25]. Altered expression of peptides as well as receptors, that belongs to this system, was observed in cancers like glioblastoma, breast hyperplasia, ovarian carcinoma, skin carcinoma, cervix carcinoma, and other [32]. Moreover, in renal cell carcinoma, changes were noticed in components of RAS like ATRs overexpression and downregulation of ACE, where alterations were related to tumor aggressiveness [33]. As well as in lung cancer alternations, gene expression of RAS components was noticed, along with its enzymes [34,35,36]. Taking into consideration the impact of RAS on cancer development, it leads to the conclusion that RAS inhibitors should block cancerogenesis. However, this is only an assumption because in practice the results of meta-analyzes vary widely. Some studies indicate an effect on survival, reduction of mortality, or recurrence-free survival [37,38,39]. In hepatocellular carcinoma, patients treated with sorafenib with an addition of RAS inhibitors resulted in improved survival [40]. On the other hand, there are many meta-analyzes, the results of which indicate no effect of angiotensin receptor blockers (ARBs) on cancer incidence, progression, or overall survival [41,42,43]. Numerous in vitro and in vivo studies indicate the potential influence of ARBs, ACE, or ACE2 on the proliferation, migration, invasiveness, or programmed death of cancer cells.

Both AT1R and AT2R are present in tumors and may be up-regulated in some of them. Ang II induces cell proliferation by activating AT1R, but stimulation of the AT2R inhibits cell growth in different cell types. The anti-proliferative and anti-angiogenic effects of the Ang 1–7/Mas axis in cancer have been evaluated. In addition, we speculate that Ang II and relaxin 2 (RLN2) are involved in the transition from the hormone-dependent to the hormone-independent phenotype via modulation of the expression of sex steroid hormone receptors.

## 2. Pituitary Gland

Ang II was found in the pituitary gland [5,6,44,45]. The main target of Ang II in pituitary are cells secreting prolactin (PRL)—lactotrophs [46]. This peptide was proved to influence the secretion of anterior pituitary hormones like prolactin (PRL), adrenocorticotropic hormone (ACTH), luteinizing hormone (LH), and growth hormone (GH) [6,47]. Moreover, Pawlikowski et al. assessed the presence of ATRs in normal rat pituitary gland, estrogen-induced rat pituitary tumor, and human pituitary adenoma [48]. This study revealed, that pituitary tumorigenesis in both, rat and human, was correlated with down-regulation of AT1R, while the presence of AT2R in the vascular walls of the intra-tumoral blood microvessels might be connected with process of tumoral angiogenesis [49]. 

Many studies revealed correlation between angiotensin and proliferation of pituitary gland cells [50,51,52,53,54,55,56,57,58]. Moreover, Ang II was observed to induce proliferation of mouse spleen lymphocytes in vitro. This effect was reversed by treatment with ATRs antagonists, losartan and PD 123319 [57] Pawlikowski et al. observed that Ang II stimulated proliferation of cells isolated from estrogen-induced pituitary tumors and human pituitary adenomas. Moreover, treatment with ACE inhibitors (ACEIs), enalapril and enalaprilate, resulted in reduced proliferation of estrogen-induced pituitary tumors. Additionally, ACEIs and AT1R and AT2R inhibitors (losartan and PD 123177, respectively) caused decreased density of PRL-immunoreactive cells and PRL serum levels in estrogen-induced pituitary tumors [50,51]. Shinkai and Ooka also observed that Ang II induced mammotroph (lactotoph) proliferation in rat pituitary. This effect was inhibited by AT2R antagonist, saralasin [59]. Not only Ang II has an impact on proliferation of pituitary gland cells, Pawlikowski and Kunert-Radek noticed that both, Ang II and Ang IV, could stimulate proliferation of these cells. The results of this study showed that Ang IV promoted [3H]-thymidine incorporation into DNA of the anterior pituitary cells. This stimulatory effect was not blocked by aforementioned inhibitors of AT1R and AT2R. Lack of inhibition of Ang II influences lactotroph by its selective inhibitor, losartan, may indicate that Ang II affects lactotroph proliferation in a different way than through AT1R [51,52]. This thesis was confirmed by Ptasinska-Wnuk et al. who observed that losartan was unable to abolish the proliferative effect of Ang II and Ang IV in the anterior pituitary of rat [53]. On the other hand, Lawnicka et al. noticed that stimulation of proliferation in rat anterior pituitary by diethylstilboestrol (DES), synthetic estrogen, was inhibited by losartan (AT1R antagonist), PD123319 (AT2R antagonist), and captopril (ACE inhibitor). This result was evidence of AT1R and AT2R involvement in estrogen-dependent mitogenic effect in rat pituitary [54]. Surprisingly, the research carried out by Ptasinska-Wnuk et al. revealed anti-proliferative consequence of angiotensins [Ang II, Ang IV, angiotensin 5-8 (Ang 5-8)] on GH3 lactosomatotroph pituitary tumor cell culture. The researchers suggested probable association with mitogen-activated protein kinase (MAPK) pathways p44/42 and p38. All aforementioned angiotensins diminished cell viability and proliferation of GH3 cells. Inhibition of MAPK p44/42, that has mitogenic and antiapoptotic effects, resulted in decrease of the inhibitory effect of Ang II, but had no influence on Ang 5-8 action. What is more, inhibition of MAPK p38, that is correlated with antiproliferative effects and promotion of apoptosis, blocked the decrease in the number of Ang II- and Ang IV-treated GH3 cells. The conclusion of this study was that MAPK p44/42 was connected with antiproliferative effect of Ang II, while MAPK p38 takes part in induction of apoptosis by Ang II and Ang 5-8 [55]. On the same cell line was evaluated the involvement of ATRs in cell growth in vitro. All tested antagonists of ATRs (losartan—AT1R antagonist; PD123319—AT2R antagonist; divalinal—AT4R antagonist) were unable to prevent the decrease of total number of the GH3 cells and their proliferation induced by angiotensins. This outcome suggests that growth inhibition induced by angiotensins is not correlated with tested ATRs [56].

Angiotensins are capable of influencing the rat pituitary cells proliferation via stimulation of tyrosine kinases (TKs) activity. TKs were correlated with transduction of growth signals [52,60]. Ang II acting through AT1R had ability to influence on activation of receptor TKs (for example platelet-derived growth factor receptor) and non-receptor TKs (for instance Src kinase family or Janus kinases) [61,62]. The study of Lachowicz-Ochedalska et al. revealed that both, Ang II and Ang IV, induced activity of TKs in estrogen-induced rat pituitary. Probably, Ang II acted on TKs activity through Ang IV, because inhibition of aminopeptidase A and aminopeptidase N (APA and APN) responsible for conversion of Ang II to Ang IV, resulted in the reduced effect of Ang II [60]. A more specific experiment concerning Ang IV influence on rat pituitary gland was carried out by Rebas et al. Treatment with Ang IV resulted in increased activity of TKs in rat anterior pituitary tissue. Addition of losartan diminished effect of Ang IV, which implies the involvement of AT1R [52]. Angiotensin III (Ang III) also could influence protein TK activity in rat pituitary in a dose dependent manner. This effect was not mediated by AT1R and AT2R because treatment with their blockers together with Ang III improved changes made by this peptide [63].

The other way of RAS effects on anterior pituitary proliferation, signal transduction, and hormone release are through inositol-1,4,5-triphosphate (IP3) and protein kinase C (PKC). Following binding of Ang II to its receptors, there is an activation of phospholipase C that results in hydrolysis of membrane phosphoinositides (IP) into inositol phosphates such as IP3. IP3 influences the release of calcium from intracellular stores, which in turn together with PKC has impact on proliferation, signal transduction, and hormone release [64,65]. Ang II influences IP3 activity [66]: Ang II significantly increased the level of IP3 [64,65,67] and PKC [64] in the rat anterior pituitary gland. The effect of different steroids on Ang II influence on the pituitary gland was examined. Treatment with estradiol improved the action of Ang II on IP3 level and PKC activity, while progesterone had no impact on the level of IP3, but changed the Ang II influence on PKC activity [64,67]. Moreover, pregnenolone sulfate was able to modify Ang II-induced alterations in the IP3 level [65].

## 3. Adrenal Gland

The local RAS is correlated with adrenal gland, its components were observed in adrenal cortex and medulla. Moreover, Ang II had an impact on aldosterone, catecholamine, and corticosterone secretion in the adrenal gland [68]. ATRs were noticed in the rat adrenal gland. AT1R was located mainly in zona glomerulosa of the adrenal cortex and also in medulla, while AT2R was observed only in medulla. Moreover, in the same study, it was observed that Ang II was present in the same part of the adrenal gland as AT1R [69]. Pawlikowski et al. carried out an experiment concerning localization of AT1R and AT2R in adrenal tumors. AT1R immunostaining was observed in adrenocortical hyperplasia and benign adrenocortical adenomas mostly in cell membranes and to a lesser extent in cytoplasm and nuclei, while in adrenal cancers and pheochromocytomas, a weak immunostaining was noticed only in cell cytoplasm. As far as AT2R is concerned, there was no observed immunostaining in all investigated tumors, however interstitial tissue and walls of intra-tumoral vessels were immune-positive for AT2R [70].

Angiotensins were proved to have an impact on adrenal gland proliferation [60,71,72]. Ang II was capable of stimulation of [3H]-thymidine uptake in rat adrenal glands. This effect was inhibited by somatostatin analog (SMS 201-995) [71]. Not only Ang II, but as well Ang IV, influenced on proliferation of rat adrenocortical cells. Treatment with Ang II and Ang IV resulted in elevated bromodeoxyuridine (BrdU) incorporation in the adrenal cortex, mainly in zona glomerulosa. Addition of losartan, AT1R blocker, inhibited only Ang II-induced proliferation. Such outcome indicates that Ang II acted through AT1R on adrenal gland proliferation, while Ang IV had another way of action [53]. On the other hand, the influence of Ang II and Ang IV on adrenocortical cell proliferation was different in ovariectomized rats. The addition of Ang II alone did not change the level of BrdU incorporation into adrenal gland cells, however, together with losartan it decreased proliferation of cells of glomerulosa and reticularis zones. Treatment with Ang IV inhibited proliferation mainly in the reticularis zone, the addition of losartan decreased this effect to some extent [72]. Peters et al. described the transgenic rat model overexpressing AT2R. They noticed that an elevated number of AT2R opposed the proliferative effect of Ang II (acting via AT1R) in the adrenal zona glomerulosa [72]. Surprisingly, Otis et al. noticed that Ang II decreased proliferation and simultaneously enhanced protein synthesis in rat glomerulosa cells. p42/p44 MAPK and p38 MAPK pathways contributed to this result [73]. Moreover, Ang II was proved to inhibit proliferation of glomerulosa cells by influencing on fibronectin-integrin signaling [74]. In terms of exocrine glands, it can be concluded that RAS plays a role in physiological processes such as proliferation or secretion of individual organic chemicals. Abnormalities in RAS are observed in both pituitary and adrenal neoplasms, mainly due to the influence of angiotensins on the increased proliferation of non-neoplastic cells. The use of both ARBs and ACEIs leads to a reduction in the action of individual angiotensins, which might serve as a potential target in cancer therapies.

## 4. Prostate Gland

In the normal prostatic epithelium, angiotensins stimulate cells proliferation. The addition of ACE inhibitor, captopril, resulted in a decreased incorporation of bromodeoxyuridine into cell nuclei (index of cell proliferation) of prostatic epithelial cells. Treatment with Ang II or Ang IV reversed the inhibition of proliferation induced by captopril. Moreover, this effect was not directed by AT1R, because addition of losartan did not block activity of angiotensin [75]. Angiotensins influence as well prostate cancer cells growth. Lawnicka et al. noticed that, in contrast to the normal prostatic epithelium, in the hormone-independent prostate cancer cell line, DU-145, angiotensin (Ang II, Ang III, Ang IV) caused concentration-dependent decrease of cell viability [55,76]. In contrary to this study, Domińska et al. observed that Ang II stimulates proliferation of the LNCaP (hormone-dependent prostate cancer) cell line, but not of the DU-145 cell line. Moreover, 24 h-incubation with Ang II resulted in inhibited cell growth. In both cell lines, treatment with Ang III caused weak increase of cell proliferation in DU-145 and LNCaP cell lines [19]. Sidorkiewicz et al. noticed that Ang II inhibited cells proliferation of the aforementioned cell line acting via AT1R. What is more, they discovered that there were two variants of AT1R in this cell line [77]. The other peptide from Ang family, Ang IV, also had effect on prostate cancer cells growth. Concerning the hormone-independent cell line, there was no significant influence of Ang IV on cell viability and proliferation, while in the case of the hormone-dependent cell line, Ang IV decreased cell growth. Moreover, the AT2R blocker (PD123319) reduced the inhibitory effect of Ang IV on proliferation of LNCaP cells, which implied that the proliferative activity of Ang IV could be mediated by AT2R. Interestingly, Ang IV was able to modulate AT1R and AT2R density in prostate cancer cells [78]. Domińska et al. reported that RAS and the relaxin family peptide system were correlated and had impact on the viability and proliferation of prostate cancer cells [79] and normal prostate epithelial cells [80]. In two prostate cancer cell lines, LNCaP (high androgen receptor activity, low invasiveness) and PC3 (low androgen receptor activity, highly metastatic), both peptides, Ang II and relaxin 2, induced proliferation, however, greater changes were observed in androgen-dependent cell line. Domińska et al. also observed a significant increase in androgen receptor expression in LNCaP cells but an insignificant decrease in PC3 [79]. In addition, treatment with peptides in combination had bigger impact than application of them separately [58]. Concerning the normal prostate epithelial cell line, PNT1A, there was similar effect of Ang II and RLN2 as in prostate cancer cell lines. Both peptides administered alone increased viability of PNT1A cells, but in combination, this effect was greater. Relating to the BrdU incorporation assay, both peptides increased proliferation, however this effect was very weak [81]. Ang II and RLN influenced proliferation and invasion of prostate epithelial and cancer cells by modulation of nuclear factor kappa-light-chain-enhancer of activated B cells (NF-kB) pathways [80]. Uemura et al. confirmed that Ang II stimulated proliferation of normal and cancer prostate cell, additionally this effect was inhibited by AT2R blockers [82].

As mentioned above, Ang II can influence cell proliferation, through modulation of TKs activity. Ang IV was capable of reducing TK activity and cell viability of the hormone-dependent prostate cancer cell line, LNCaP. Ang IV acted, at least partially, through AT2R, because addition of its blocker, PD123319, decreased inhibitory effect [83]. Domińska et al. observed that steroid hormones, testosterone, and 17β-estradiol were capable of reversion of the action of Ang II and Ang 1-7, which decreased protein TK activity in the late-stage prostate cancer cell line—DU145 [84]. Ito et al. noticed that inhibition of AT2R by its agonist, Compound 21 (C21), resulted in reduction of proliferation of prostate cancer cells and transgenic rat for adenocarcinoma of prostate (TRAP). Moreover, C21 decreased expression of androgen receptor and activity of prostate-specific antigen (PSA) promoter [85]. Woo et al. reported that AT2R blockers inhibited proliferation of prostate cancer cells. Moreover, in this study was observed that those blockers induced autophagy-associated cells death [86].

In prostate gland AT1R and AT2R, stromal and epithelial structures were found. Similarly, to endometrial cancer, the highest expression of these receptors was correlated with Gleason 2 grade (well differentiated) in neoplastic epithelium in comparison to non-neoplastic epithelium and Gleason 3-5 grade [49]. Moreover, the expression level of AT1R and AT2Rs in the rat prostate on protein and mRNA level was correlated with myocardial infarction [87]. Ang II and relaxin 2 can modulate expression of androgen receptors (ARs) in prostate cancer cells.

## 5. Endometrium

Angiotensins were involved in regulation of proliferation of rat endometrium [75] and human endometrial cancer cells [88]. Ang II and Ang IV induced proliferation of rat uterine epithelium. Addition of losartan had no impact on Ang IV-induced proliferation, while administered together with Ang II also increased BrdU incorporation into cell nuclei in a bigger degree than Ang II alone [75]. Ang II increased viability of human endometrial cancer cell lines (Ishiwaka, MFE296 and MFE280) [88]. Shan et al. noticed that Ang 1-7 was capable of inhibition of Ang II-induced proliferation of human endometrial stromal cells [89]. Blockage of AT1R by telmisartan caused inhibition of proliferation of endometrial cancer cells and tumor growth in nude mice [90].

The expression of ATRs was assessed in endometrial [91,92] and prostate cancer [49]. Delforce et al. noticed that mRNA of AT1R, ACE, and (pro)rennin receptors were highly expressed in endometrioid carcinomas and their adjacent endometrium, which suggested their role in development of endometrial cancer [93]. In uterine corpus cancer, a correlation between cancer stage and receptor expression was noticed. Piastowska-Ciesielska et al. showed that the highest level of AT1R was in the well differentiated (G1) stage, while the lowest was observed in the poorly differentiated (G3) stage. Probable explanation of AT1R dominance in the G1 stage in comparison to higher stages was involvement of Ang in the process of angiogenesis occurring on the beginning of cancer development [91].

The presence of RAS components is observed in both healthy and malignant organs of the genitourinary system. Angiotensins stimulate cells proliferation in both the prostatic epithelium and rat endometrium. Additionally, Ang II increases the vitality of human endometrial cancer cell lines. The use of inhibitors such as telmisartan led to an inhibition of the proliferation of endometrial cancer cells and tumor growth in nude mice. The use of individual angiotensins is not limited only to proliferation, Ang IV leads to modulation of the AT1R and AT2R density. The information presented above shows the wide spectrum of the involvement of RAS in a multitude of processes. Nevertheless, the endometrium is one of the topics that requires better understanding.

## 6. Other Organs/Cancers

### 6.1. Brain

As previously mentioned, the role of the RAS has two faces, pathological and physiological without which the proper functioning of the organism would not be possible. Scientific evidence suggests that the effects of angiotensins on the brain vary widely [18,22,31,94,95,96]. Highest concentrations of AT1R are located in brain areas important in the regulation of body-fluid balance and blood-pressure maintenance. A perfect example of the positive effects of angiotensin on body function is the regulation (decreasing) of nocturnal blood pressure in rats. The potential use of such information is in the case of patients being treated for hypertension. It was indicated that night-time ambulatory BP (blood pressure) is more closely associated with fatal and nonfatal cardiovascular events (stroke, myocardial infarction, and cardiovascular death) than daytime ambulatory BP. Additionally, to confirm the presence of RAS elements, is the infusion of Ang II into the brain which increases BP and central injection of purified Ang II near the hypothalamus that resulted in a drinking response [97,98,99]. The degree of influence of the RAS on, the functions of the brain are also immensely interesting. The use of AT1R antagonists demonstrated anxiolytic- and antidepressant-like effects. The anxiogenic effect caused by Ang II administration was partially mitigated by treatment with the AT2R antagonist [30,95]. The role of the AT2R in maintaining adequate behavioral responses is also significant. Mice with AT2R deficiency presented increased anxiety-like behaviors [96]. In drawing a conclusion from the presented data, the pattern of interaction and balance between AT1R and AT2R is highly similar to those presented in other organs. Maintaining the balance between the activation of the AT1R and AT2R allows for an adequate response of the organism to external stimulus, in the form of depressive-like behaviors and anxiolytic-like effects, respectively. The destructive effect of RAS overactivity, apart from depression, is also observed in neurodegenerative diseases such as Alzheimer’s disease [100,101,102]. The potential role of RAS, and more precisely ACE, is attributed to the potential break down of β-amyloid (Aβ) peptides, thereby inhibiting the formation of amyloid plaques [103,104]. Additionally, an ACE analog, ACE2, which acts in the context of Aβpeptides degradation has a catalytic role in the cleavage of β-amyloid peptide 43 (Aβ43) to β-amyloid peptide 40 (Aβ40). Moreover, the use of AT1 antagonists and ACEIs gives promising results by reducing the frequency and/or severity of migraine attacks [105,106,107].

AT4R subtype, which was originally discovered in the brain as a binding site for Ang IV ligands, facilitates learning and memory process [15,22]. The distribution of AT4R and with the cholinergic neurons and their projections, particularly in the septum, hippocampus, and cortex, largely overlaps [108]. The overlapping of AT4Rs and cholinergic neurons suggests participation in the process of learning and memory. In addition, AT4R ligands could be the facilitation of glucose uptake into neurons, cognitive function correlates with enhanced glucose utilization [15].

### 6.2. Breast Cancer

In operable breast cancer, expression of AT1R was correlated with increased cell proliferation [109]. Ang II increases the proliferation of breast cells and influences angiogenesis via AT1R in invasive ductal breast cancer [110]. Lewandowska et al. carried out research that assessed the influence of Ang II on TK activity in two breast cell lines—MCF-7 (hormone-dependent) and MDA-MB-231 (hormone-independent). Treatment with Ang II resulted in reduced activity of TKs in the MDA-MB-231 cell line, while in the MCF-7 cell line, this effect was similar but very weak. Addition of estradiol had no effect on the hormone-independent cell line, but in the case of the estrogen-dependent cell line, this hormone reduced TK activity much more than Ang II alone. Interestingly, real-time quantitative polymerase chain reaction (RT-qPCR) PCR revealed presence of both, AT1R and AT2R products in MDA-MB-231 cell line, while in MCF-7, there was expression only of AT1R. Since AT2R was correlated with apoptosis and was capable of inhibiting the function of AT1R, it can be explained why in the MDA-MB-231 cell line, which possess AT2R, Ang II through binding to this receptor reduced cell viability and TK activity [111]. Oh et al. observed that in the xenograft model of breast cancer overexpression of AT1R was correlated with acceleration of tumor growth and increased expression of proliferation marker protein Ki67 (MKI67) [2]. In ovarian cancer, AT2R-interacting protein 3a inhibited proliferation of cells [112].

### 6.3. Liver

RAS also plays a significant role in the pathophysiology of the liver. The neoplastic process that leads to the occurrence of hepatocellular carcinoma (HCC) is a complex process and is preceded by two basic pathophysiological conditions: liver inflammation and cirrhosis. The factors leading to hepatitis are mainly hepatitis C virus (HCV), hepatitis B virus (HBV), or alcohol and others [113]. Each of the above-mentioned steps are closely related to angiogenesis. Treatment with thioacetamide (TAA) in rat models induced liver fibrosis [114]. Ang II was reported to induce the expression of tumor growth factor β1 (TGF-β1) [114]. The use of perindopril, an ACEI suppressed tumor necrosis factor α (TNF-α) expression, as compared with TAA group. Moreover, perindopril administration resulted in a limitation of smooth muscle actin alpha 2 (α-SMA) expression to the portal areas as well as markedly downregulated the expression of AT1R in liver cells and stellate cells [114,115]. In cholangiocarcinoma [116] and hepatocellular carcinoma [117], inhibition of AT1R by telmisartan also resulted in blockage of proliferation. Additionally, it was noticed that telmisartan induced G0/G1 cell cycle arrest by inhibition of G0 to G1 cell cycle transition. However, the use of ACEIs or ARBs showed that no significant difference was observed in HCC risk within 7 years between the initial exposure and non-exposure groups (HBV and HCV cohorts) [118]. Moreover, there was an observed decrease in the cell cycle-related proteins [116,117]. Taken together, it might be assumed that the RAS role in the pathogenesis of liver diseases is also present, however not fully revealed.

### 6.4. Lung

Given the current epidemiological situation, the vast majority of scientific data are based on the severe acute respiratory syndrome coronavirus 2 (SARS-CoV-2) topic. Nevertheless, the role of RAS is not only based on the relationship with coronavirus disease 2019 (COVID-19), but also participates in the process of physiological functioning, proliferation, migration, and the overall process of oncogenesis [14,34,35,119,120,121]. The use of AT2R antagonists affects many aspects of neoplastic cells. Apoptosis rate in the telmisartan-treated group (A549 cell line) was significantly higher than that in the control group, which is also essential that telmisartan had a greater effect on early apoptosis compared with late apoptosis. This effect is directly related to the increased expression of pro-apoptotic proteins such as caspase-3 and B-cell lymphoma 2 (Bcl-2) associated X, apoptosis regulator (Bax) [121]. In addition, expression of anti-apoptotic protein Blc-2 decreased. An analogous situation is observed in the case of matrix metalloproteinases, the expression of genes encoding matrix metalloproteinase-2 (MMP-2) and matrix metalloproteinase-9 (MMP-9) is not directly decreased, and catalytic activity is not reduced. The reduction of the catalytic activity of metalloproteinases affects cancer cells invasion and migration. The increase in MMP-2 and MMP-9 mRNA concentrations was observed when AngII was applied to the A549 cell line [35,121,122].

Both in vitro and in vivo, Ang 1-7 reduces proliferation of lung cancer cells (A549, Spc-A1 and Spc-A1 xenograft tumor cell, respectively) [120]. One of the potential elements by which Ang 1-7 reduces the growth of neoplastic cells is the reduction in cyclooxygenase-2 (COX-2) activity, an enzyme whose pro-inflammatory, pro-angiogenic, and tumor growth acceleration properties control the tumor microenvironment (TME). In in vivo studies, the control group was infected with a vector targeting the expression of Ang 1-7 (AAV-Ang-(1-7)-treated group) the volume and mass of tumor were reduced compared to the control group [120]. Moreover, there is also a reduction in the mRNA concentration of the cell division control protein 6 homolog (Cdc6) molecule, involved in the epithelial–mesenchymal transition (EMT) process. The Mas receptor participates in this process, due to the fact that after the use of a selective inhibitor, the migration properties were recovered.

The alternative axis of the RAS-ACE2/Ang 1-7/Mas deserves special attention to the presence of the axis significantly influences the functioning of cancer cells both in vitro and in vivo. Anti-migratory and anti-invasion effect of Ang 1-7 was mediated through inactivation of the phosphoinositide 3-kinase (PI3K/Akt), p38 MAPK, and c-Jun N-terminal kinases (JNK) signaling pathways [120]. Ang 1-7 has no effect on the expression of genes encoding PI3K/Akt and MAPK, and thus the protein concentration in cells is not changed. The changes mainly concern the degree of phosphorylation PI3K/Akt and p38 MAPK [36,120]. An example are the results presented by Chen et al. [120]. The use of Ang 1-7 resulted in a reduction of BrdU positive cells in the test group (SPc-a1, A549) compared to the control group, thus DNA synthesis was reduced. The use of a selective inhibitor for the Ang1-7 receptor eliminated the above-mentioned effects; therefore, it can be concluded that these effects are induced by activation of the Mas receptor.

### 6.5. Ocular Disorders

The role of RAS in physiological and pathological processes is of increasing interest. The ocular RAS deserves special attention. Apart from the standard, physiological role of RAS, as in most of the cases presented in this study, it also plays a significant role in pathophysiological processes [123,124,125,126]. All components of the RAS are currently presented on the retina, other elements of the eye, such as iris, cornea, also show the presence of some RAS elements [29]. Disease entities related to the organ of vision occurs mainly in the area of the retina we distinguish as follows; diabetic retinopathy (DR), age-related macular degeneration (AMD), glaucoma, retinitis pigmentosa [127,128,129]. The appearance of the components of the RAS was originally observed in the eye, but the initial one could not clearly state whether these are elements delivered to the eye with blood or were locally synthesized in the eye. Confirmation of the presence of RAS ocular and the statement that angiotensinogen, Ang I, and Ang II from plasma could not enter into the eye allow the conclusion that RAS has an imperative role in the ocular pathophysiology [130,131]. Concentration of Ang I and II were 5- to 100-fold higher than could be accounted for by contamination with blood, which excludes significant influence of circulating RAS on ocular pathophysiology [130]. In addition to the standard activation of AT1R and AT2R, the activity of which is covered in earlier sections of this publication, an alternative, Ang II independent activation via the (pro) renin receptor [(P)RR] deserves special attention. (P)RR binds to both renin and pro-renin without conventional catalytic cleavage of prorenin [132]. The end effect associated with the activation of (P)RR is highly identical to that exhibited during activation of the AT1R [132,133]. An example is diabetic retinopathy where the activation of AT1R and (P) RR leads to the acceleration of the inflammatory process and angiogenesis. The use of both AT1 and (P)RR receptor blockers allow to achieve a protective effect by reducing the degree of inflammation, oxidative stress, or reducing angiogenesis by inhibiting signaling through the ERK pathway. The presence of ACE that converts Ang to specific target forms also plays a critical role, the use of ACEI resulted in a reduced expression of vascular endothelial growth factor (VEGF)—as previously mentioned, responsible for the angiogenesis process [123,124,125,126,127,128,129,131,132,133,134,135,136,137,138,139,140,141,142,143]. A similar example concerns AMD, the degradation in the eye is mainly caused by the new formation of blood vessels (neovascularization), the use of ACEIs, AT1RBs, and (P)RRBs reduced the inflammatory process, thus inhibiting the progression of choroidal neo-vascularization [28,144,145,146,147,148]. The nature of glaucoma is slightly different, the important thing in glaucoma is intra ocular pressure. Management in treatment is aimed at reducing intraocular pressure (IOP). AT1R signaling regulates aqueous humor formation, secretion, uveoscleral outflow, and IOP [29]. The use of both ACE inhibitors, AT1R blockers such as mi Ang 1-7 leads to the achievement of therapeutic effects: reduction of aqueous humor formation and increasing uveoscleral outflow, and reduced IOP [149,150,151,152,153,154,155,156,157].

### 6.6. Colon Cancer

The versatility of the RAS also shows a role in the physiological and pathological processes of the gut [158]. As in the previously presented cases, abnormalities in the functioning of RAS lead to pathophysiological conditions which also include neoplasm [159]. In the context of in vitro studies, both tumor colon cancer cell lines (HT29 and CT26) express the AT1R, but do not express angiotensinogen and ACE [159]. A possible alternative to the absence of ACE is chymase (where its presence in the alternative pathway enables catalytic transformation of angiotensin I to angiotensin II) [160]. An interesting point is that in the case of colorectal cancer is that Ang II concentration in the primary tumors was higher in patients with liver metastasis than that in those without liver metastasis. The use of angiotensinogen on the HT29 cell line provided significantly pronounced effects of growth, invasion, and apoptosis inhibition, but Ang was unable to produce these changes in CT26 cells [158]. Ang II enhanced cell growth and in vitro invasion into type IV collagen in the HT29 and CT26 in dose-dependent manner was observed. In contrast, results were observed with Ang II in the context of apoptosis—apoptosis was decreased, also in a dose-dependent manner. Additionally, it was proved that high blood sugar induced Ang II activation in colorectal cancer cells (CRC), which may be of potential importance in the case of diabetic patients. This presents a potential role of diabetic status on CRC progression [161]. Irbesartan, an ARB, restores the degree of ACE2 expression in the study group (stress-induced intestinal inflammation diminished ACE2 expression in intestinal mucosal surface), what is more, there is no change in ACE or ACE2 expression in the group without the induced stress that in treatment group resulting in intestinal inflammation. Additionally, irbesartan administration suppressed stress-induced increase in ACE and angiotensinogen mRNA [161]. Stress resulting in intestinal inflammation also led to toll like receptor 4 (TLR-4)-induced mRNA expression, followed by significant increase in interleukin 1β (IL-1β) in plasma and intestine, an effect that was mitigated after irbesartan treatment. The use of ARB also reduces the severity of colitis by reducing colonic inflammatory cell infiltration [162]. Studies on ACE activity in intestinal cancer cells (HT29) showed a significant increase in the activity of this enzyme. In the tested material, there was also an increased expression of genes encoding such molecules as c-reactive protein (CRP) (15-fold increase, *p* < 0.05)) and inducible nitric oxide synthase (iNOS) (4-fold increase, *p* > 0.05). CRP and iNOS are considered as inflammation-related molecules in tumors [20,163]. In the context of in vivo studies, the use of losartan and enalapril in a duet did not lead to additional treatment benefits in the case of dextran sulfate sodium (DSS)-induced colitis in mice. Alone, enalapril treatment resulted in a lowering of the overall histopathologic scores and IL-1β expression, losartan alone reduced the macroscopic and histopathologic inflammation scores.

In summary, RAS plays a key role in both the physiological and pathophysiological processes taking place in the abovementioned organs, from proliferation, inflammation, learning to hematopoietic stem cell (HSC) differentiation, which lead to liver fibrosis. An analysis of the relationships between individual organs/tissues will allow a better understanding of the influence of RAS on the occurrence of pathological conditions. The enormous scope of RAS activity results, first of all, from the universality of action, which manifests itself through the activation of specific pathways in target cells. The collected data clearly show the diversity of the RAS and allow us to look at the RAS not only as the system responsible for modulating the functioning of the cardiovascular system. The rest of the analyzed information on the impact of RAS on physiological and pathophysiological phenomena is presented in Table 1.

## 7. Angiogenesis

The RAS was involved in new blood vessels formation during cancer development and in normal conditions. The mechanism of action of Ang II concerns mainly activation of VEGF in the physiological and pathological conditions [184]. Ang II and AT1R signaling had influence on tumor blood supply acting via different mechanisms [185]. Furthermore, there are also scientific data that clearly show a correlation between AT1R and vascular endothelial growth factor (VEGF) in an early stage of differentiation of endometrial cancer [91]. Ang II was found to stimulate in vitro VEGF secretion in human vascular smooth muscle cells, human mesangial cells, rat heart ECs, and bovine retinal microcapillary pericytes [186,187,188,189].

Ptasinska-Wnuk et al. studied the influence of Ang peptides (Ang II, Ang III, and Ang IV) on angiogenesis in GH3 cell line and primary culture of rat PRL-secreting DES-induced tumor. They observed that all aforementioned peptides increased VEGF concentration in GH3 cells and rat adenoma cells. Moreover, this effect was mediated by ATRs in GH3 culture, while the stimulatory effect on VEGF by Ang II was decreased by administration of AT1R and AT2R antagonists, losartan and PD123319, respectively [54,190]. Similar effect was noticed by Lawnicka et al. in the rat anterior pituitary. Treatment with DES caused elevated VEGF reactivity in blood vessels and anterior pituitary cells. Addition of AT1R and AT2R antagonists and ACE inhibitor caused diminished the number of VEGF-positive blood vessels [54]. Those outcomes suggested that RAS takes part in stimulation of angiogenesis by induction of VEGF secretion acting by AT1R and AT2R in estrogen-induced adenoma [56]. In another study, the influence of Ang IV on VEGF secretion in GH3 cells was additionally tested. Low concentrations of Ang II, Ang III, and Ang IV increased VEGF level, but on the other hand, higher concentrations of Ang III and Ang IV resulted in inhibition of VEGF secretion [58]. The same process occurs with Ang II where doses of 10^−6^ M and higher lead to inhibit VEGF secretion in primary rat prolactinoma culture. It perfectly shows that the action of angiotensin is dose-dependent [190]. There are also data obtained from in vitro studies on multidrug resistant A549/cisplatin (A549/DDP) cells that indicate that ACE2 leads to a reduction of both mRNA and protein products for ACE, AT1R, and VEGF [36].

Arrieta et al. observed that expression of AT1R in operable breast cancer was related to high vascular density, which suggests enhancement of angiogenesis [109]. Similarly, in the breast cancer cell line, MCF-7, overexpression of AT1R resulted in induction of angiogenesis. This effect was suppressed by AT1R antagonist, losartan. On the other hand, in the MDA-MB-231 cell line (breast cancer cells), Ang 1-7 diminished Ang II-triggered pro-angiogenic mechanisms like increase of VEGF expression [191]. In terms of in vivo studies, VEGF mRNA and protein levels were significantly reduced in tumors from nude mice where viral vector–mediated expression of Ang (1-7) (AAV8 (Y733F)-CBA-Ang (1-7)) was performed [192]. Furthermore, usage of viral vector-mediated expression of Ang (1-7) may also attenuate VEGF and placental growth factor (PlGF) signaling by reducing the number of available receptors (VEGF receptor Flt-1 and Flk-1). Hence, suggesting that decreases in VEGF and PlGF receptors in response to Ang (1-7) contribute to decreases in cell and tumor growth both in vivo and in vitro [193]. Chen et al. reported that Ang II promoted expression of vascular endothelial growth factor A (VEGF-A) in MCF-7 cell line. This effect was also inhibited by addition of losartan [2]. Moreover, in mice with MCF-7 xenografts, treatment with candesartan resulted in inhibition of tumor growth and angiogenesis [193]. It was observed that induced overexpression of this heptapeptide caused reduction of vessel density [194].

In endometrial adenocarcinoma, the correlation between expression of VEGF, AT1R, and AT2R and grade of differentiation was observed. VEGF was related to later invasion of the myometrium, because the VEGF mRNA expression level was significantly higher in G3 than in G2. Moreover, there was significant correlation between AT2R and VEGF expression, while in early stage of endometrial cancer, there was inverse correlation between AT1R and VEGF [91,92]. Piastowska-Ciesielska et al. reported that Ang could influence VEGF via the kinase domain-containing receptor (KDR), the major VEGF receptor in human cancers. They noticed that there was significant association between KDR and AT1R expression on protein and mRNA level. This outcome suggested that Ang II acting through AT1R influenced the VEGF level via KDR expression in endometrial cancer [92,194]. Dobrzycka et al. observed that in women with endometrial cancer (type I and II), there was a higher level of VEGF than in healthy patients [92,195].

ACE2 is responsible for conversion of Ang II to Ang 1-7. In non-small-cell lung carcinoma (NSCLC), overexpression of this enzyme resulted in suppression of VEGF expression [36]. Disturbances in VEGF concentration in cancer cells as mentioned above are in part due to individual components of the RAS. VEGF produced by tumor cells has, inter alia, autocrine activity. Consequently, binding to receptors on the surface of their own cells leads to a situation where increased VEGF production improves the angiogenesis process [196,197,198]. The proposed mechanism is related to the overexpression of mRNA and (kinase domain-containing receptor) KDR protein itself, the concentration of which is significantly present in lung and ovarian cancers [198,199,200,201].

### 7.1. Angiotensins Have an Impact on Invasion and Metastasis

It is known, that one of the stages of cancer development are invasion and metastasis to distant places, which could be related to increased mobility of cells and alterations in their attachment to the extracellular matrix (ECM) and other cells [201]. 

The RAS played a role in cancer metastasis by modulating adhesion, migration, and invasion. Treatment with ACEIs and ARBs may be beneficial against cancer metastasis [202]. RAS is correlated with colorectal cancer liver metastases [203]. Treatment with Ang II resulted in promotion of cellular migration and reduction of adhesion of melanoma cells [203], moreover, inhibition of AT1R by losartan resulted in reduction of migratory activity, while AT2R inhibition by PD123319 caused an increase of adhesion and invasion [204]. A similar effect was observed by Zhou et al. in colorectal carcinoma. Ang II acting via AT1R was correlated with invasion, while AT2R was related to inhibition of invasion [205]. Inhibition of AT1R in non-small-cell lung carcinoma (NSCLC) by telmisartan [121] and in pancreatic cancer by miR-410 [206] also resulted in reduction of cellular invasion and migration [121]. Additionally, in hepatocellular carcinoma, overexpression of AT1R was related with promotion of invasion and migration [207]. Frequently, there were observed changes in E-cadherin (CDH1) expression, the main adhesion molecule [201]. Ang II influenced the CDH1 level in endometrial cancer cell lines, Ishikawa and MFE280 [88]. Domińska et al. reported that Ang II and Ang IV were capable of modulation of cell migration in prostate cancer. Treatment with Ang III resulted in increased migration of the LNCaP cell line in greater degree than in the DU-145 cell line, while Ang II had influence only on the DU-145 cell line. Since DU-145 was aggressive and in the tumorigenic cell line, the result of the study suggested that Ang II was correlated with stimulation of migration of metastatic cancer cells, while Ang III was related more with invasion of nonmetastatic cells in prostate cancer [19]. Interestingly, addition of Ang II stimulated adhesion of PC3 and LNCaP cells to ECM proteins, collagen I and IV, fibronectin, and laminin. Moreover, the Transwell invasion assay revealed that Ang II induced invasiveness of both prostate cancer cell lines. Treatment with Ang II increased secretion of metallopeptidases (MMPs), MMP-2 and MMP-9, that were responsible for degradation of basement membrane proteins, which was related to invasion and metastasis [79]. In the normal prostate epithelial cell line, PNT1A, Ang II increased adhesion to ECM proteins and cell migration. Contrary to prostate cancer cell lines, secretion of MMPs was decreased by Ang II [81]. In mammary epithelial cells, the 184A1 cell line, Ang II also influenced on cell motility acting via AT1R. Adhesion of 184A1 cells to fibronectin was increased after Ang II treatment. Moreover, migration through the Boyden chamber and ability to wound closure in the wound healing assay were also stimulated by Ang II. This peptide was able to induce expression of MMP-2 in 184A1 cells. All mentioned effects of Ang II on cell motility of mammary epithelial cells were inhibited by addition of AT1R antagonists [208]. In endometrial cancer cell lines (Ishikawa, MFE296 and MFE280), Ang II in different concentrations was also capable of stimulation of cell migration in the wound healing assay. As far as adhesion was concerned, the way of action of Ang II was dependent on the degree of differentiation of cell lines. In MFE280, the poorly differentiated cell line, Ang II reduced adhesion to ECM proteins; in MFE296, moderately differentiated, there was no significant change; while in the well differentiated cell line, Ishikawa, treatment with Ang II induced adhesion to ECM proteins [88].

In HepG2 cells (hepatocellular carcinoma cell line), Ang II enhanced migration and invasion. What is more, excessive expression of AT1R induced progression of hepatocellular carcinoma, while overexpression of AT2R resulted in the opposite effect [209]. Ping et al. noticed similar effect of AT2R in ovarian carcinoma. AT2R-interacting protein 3a blocked ovarian carcinoma migration and invasion by regulation of the ERK/EMT pathway [112]. In colorectal cancer, Ang II enhanced cell migration, however this effect was suppressed by AT1R and AT2R inhibitors [210].

In breast cancer cell lines, Ang II increased metastases in vivo and induced cell adhesion and migration [211]. The study of Cambados et al. showed that in the breast cancer cell line, Ang1-7 diminished Ang II-induced migration and invasion. Moreover, this peptide had influence on decrease in MMP-9 expression and activity, which were induced by Ang II [191]. Similar outcomes were observed in the study concerning head and neck squamous cell carcinoma (HNSCC). Hinsley et al. observed that Ang II, acted via AT1R, stimulated migration and invasion of HNSCC, while Ang1-7 blocked Ang II-induced cell motility [212].

Metastatic potential of prostate cancer can be related to caveolin-1 (CAV1), member of caveolin family of proteins. This protein has a dual role in cancerogenesis, sometimes it inhibits cancer development, but typically it is correlated to malignant progression. Piastowska-Ciesielska et al. reported that Ang II induced higher expression of CAV1 on mRNA and protein level in the PC3 cell line (human metastatic prostate adenocarcinoma cell line), while treatment with candesartan (AT1R blocker) resulted in decreased expression of CAV1 on both levels [179]. Kowalska et al. noticed that co-expression of CAV1, AT1R, and forkhead box M1 (FOXM1) had influence on metastatic properties in breast and prostate normal and cancer cell lines. FOXM1, a transcription factor, which elevated expression, was observed in various cancers, contributed to invasion and metastasis. In normal breast and prostate cell lines, there were high expression levels of investigated genes, which suggests their involvement in proper cell functioning. Concerning cancer cell lines, those genes seemed to influence the metastatic potential of breast and prostate cancer cell lines. AT1R probably took part in the CAV1 and FOXM1 signaling pathway in those cancer cells [213].

### 7.2. Epithelial-to-Mesenchymal Transition (EMT)

Invasion and metastasis strongly correlate to EMT. This process concerned loss of epithelial features like cell–cell contact or changes in cell polarity and gaining the mesenchymal characteristics such as motility (migratory and invasive properties). EMT leads to increase of cell motility from epithelial tissue and acquires the ability to create metastases in distant sites [191,214]. 

Ang II was capable of induction of EMT in the non-tumorigenic mammary epithelial cell line (NMuMG cell line), treatment with Ang II resulted in acquisition of the mesenchymal-like phenotype. Moreover, those changes were observed on the level of gene expression. Epithelial markers (like E-cadherin and β-catenin) were blocked (loss of E-cadherin) by Ang II, while mesenchymal markers (such as fibronectin) were overexpressed. Literature data also showed the possibility of replacing individual cadherin switching to other cadherins (e.g., N-cadherin replaces E-cadherin), or delocalization of junctional E-cadherin [166]. The role of the ACE/ACE2 balance is also not indifferent. ACE2 overexpression induced by diminazene aceturate (DIZE) lead to inhibition of the EMT induced by silica [14]. Interestingly, Ang 1-7 added together with Ang II diminished Ang II-induced changes in EMT markers expression [191]. Concerning NSCLC, Ang 1-7 suppressed EMT process by lowering Cdc6 expression [120]. Moreover, after Ang 1-7 treatment, the level of CDH1 was elevated [194]. In breast cancer cell line, MCF-7, Ang II acting via AT1R influenced on EMT. Overexpression of this receptor led to changes in EMT markers expression, reduction of CDH1, loss of E-cadherin, increase in N-cadherin expression, as well as increased EMT-related transcription factors such as phospho-Smad family member 3 (Smad3), smad-family member 4 (Smad4), and snail family transcriptional repressor 2 (Snail 2). This effect was suppressed by addition of losartan, AT1R inhibitor [2]. Additionally, in hepatocellular carcinoma, overexpression of AT1R caused promotion of EMT. In colorectal carcinoma, AT1R was correlated with promotion of EMT, because inhibition of AT1R resulted in increase of CDH1 and decrease of ZEB1 and vimentin (mesenchymal markers), while blockage of AT2R caused decrease of CDH1 [210]. The same results were observed for lung cancer lines (A549) where treatment with TGF-β1 significantly decreased the gene expression of CDH1 (encoding E-cadherin), increased the gene expression of ACTA2 (encoding α-SMA) and VIM (encoding vimentin) [215]. The above-mentioned changes are indicators of the presence of EMT, moreover, the pre-treatment using Ang (1-7) effectively eliminated these changes. Use of a selective AT1R antagonist (telmisartan) caused negative effect on the migration ability of A549 cells. The acceleration of migration and invasiveness is therefore dictated by the activation of the AT1 receptor. These results indicate that telmisartan significantly inhibits invasion and migration of A549 cells [121].

In intrahepatic cholangiocarcinoma (ICC), addition of Ang II resulted in alterations in expression of EMT markers, decrease of CDH1 and increase of vimentin expression level [166].

## 8. Discussion

Summarizing all the information presented in this review, RAS plays a critical role in both physiological and pathophysiological processes in many organs and at various stages of life (Figure 1). RAS is a potential target of many therapies: anti-cancer, treatment of progressive eye diseases, or in neurodegenerative diseases. In addition, the field of science where the role of RAS and the use of ARBs and ACEIs can be potentially implemented is a resource of diseases based on the destructive effect of inflammation. An interesting and currently poorly understood area is the influence of RAS on the functioning of the brain in the context of behavior, response to external stimuli, or the influence of the renin-angiotensin system on the occurrence of psychiatric diseases. The main limitation is the fact that the vast majority of these results are obtained in vitro or in vivo using appropriate models. It is difficult to say unequivocally that the data obtained from in vitro studies can be directly related to the potential effect in human therapy. Some of the presented data, despite the positive impact of ARBs or ACEIs, does not meet the statistical significance requirement. An example of this is many works in the field of meta-analyzes which clearly show that the use of ARBs or ACEIs does not affect the incidence of neoplastic diseases or the improvement of the patient’s condition in specific disease entities. The amount of data available in the databases is enormous. Summarizing these data on the potential use of ARBs and/or ACEIs in neoplastic diseases may form the basis for a separate review. Creating such a summary will certainly help to understand the potential use of RAS inhibitors in a simpler way. In the future, targeting specific elements of RAS may allow to obtain a therapeutic effect as the main therapy or an element supporting the main therapy in order to improve its efficiency. The combination of knowledge that allows for early detection of the stages of neurodegenerative diseases with knowledge about the influence of individual RAS elements and their impact on the initiation and/or progression of these diseases can potentially critically affect the quality of life of patients with the stigma of predisposition to neurogenerative diseases and comorbidities. A deeper understanding of the mechanisms that regulate or are regulated by RAS will allow, above all, a wider application of the acquired knowledge, but also to translate it in a more effective way into potential therapies in many human disease entities. Thus, giving a chance to treat patients in whom, due to the current health state, treatment is not effective or the use of the standard treatment regimen is not possible for various reasons.

## Figures and Tables

**Figure 1 cells-10-00381-f001:**
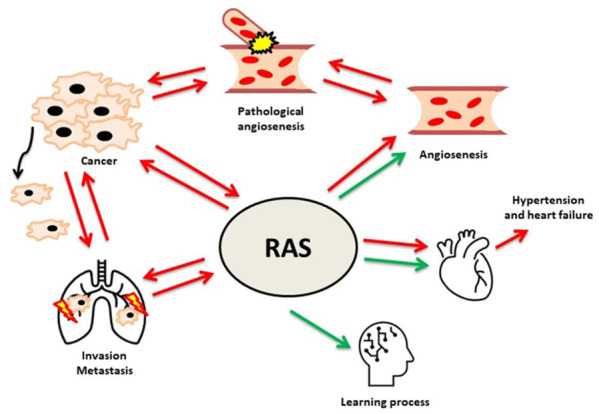
The various effects of the renin-angiotensin-system (RAS).

**Table 1 cells-10-00381-t001:** Other organs/medical conditions and potential influence of renin-angiotensin system (RAS) on their function and/or occurrence.

Organs	Data
Kidney	•Exercise training attenuates the progression of glomerular sclerosis and renal interstitial fibrosis in chronic renal failure rats by increasing expression of RAS components such as angiotensinogen and angiotensin-converting enzyme (ACE) [164]. •Increase in AT2R expression after estrogen treatment in the mouse kidney [165]. •Estrogen treatment in OVX mice dramatically decreased the AT1R to AT2Rratio by upregulation of AT2R expression [165]. •Ang II actively participates in renal fibrosis and in the parts mediated by TGF- β [166]. •Irbesartan reduces the expression of TGF-β1 mRNA [166]. •In the rat kidney, the distribution of the AT4R was reported to occur in high levels in the proximal tubules [167]. •Infusion of Ang IV into the renal artery of rats resulted in increased renal cortical blood flow and urinary sodium excretion [168]. •AT2R mRNA has been reported to have a widespread distribution within the rat kidney [169].
Gallbladder	•ACE2 suppressed tumor growth in gallbladder cancer [170]. •Lowered ACE2 expression was correlated with larger tumor size, high TNM stage, lymph node metastasis, and invasion in squamous cell/adenosquamous carcinoma patients [171].
Heart	•Ang IV stimulated DNA and RNA synthesis in cultured rabbit cardiac fibroblast [172]. •The activation of ERK1/2 was critical for the growth-promoting actions of ang II in cardiac fibroblasts or prostate cancer cell subcultures [173]. •Enhanced vasoconstrictive effect of Ang II in AT2R-knockout mice [174]. •Vasodilatation due to AT2R overexpression in vascular smooth muscle cells [175]. •Ang IV stimulates protein synthesis in rabbit cardiac fibroblasts [172]. •Ang (1-7) treatment leads to decrease the ratio of expression of MMPs/TIMPs in human cardiocytes [176]. •Ang II induces SIF complex formation in neonatal rat cardiac myocytes in a time- and dose-dependent manner [177].
Muscle	•Ang II stimulated angiogenesis in the rat cremaster muscle [178]. •Janus tyrosine kinase-STAT pathway directly through the AT1R in smooth muscle cells and cardiac myocytes [177]. •Cav-1 plays a critical role in the key signaling step in which angiotensin II induces the transactivation of the epidermal growth factor receptor (EGFR), leading to the hypertrophy and migration of vascular smooth muscle cells [179]. •Blockade of AT1R signaling reduced tumor growth, angiogenesis, and metastasis in the model of murine sarcoma and fibrosarcoma cells [180].
Stomach	•The local production of Ang II in gastric cancer has been indicated to promote lymph node metastasis and cancer progression [180]. •Increased OS median in ACEI/ARB group compared to the non-ACEI /ARB group [181].
Testicles	•Ang III and IV, and ACE2 may serve as the regulators of testicular steroidogenesis [182]. •Ang 1-7 can bind to a third receptor, specific to Ang 1-7 only (AT1-7/Mas receptor), which is also found in testes of fertile and infertile men [12]. •Main localization of Ang (1–7) and Mas receptor is observed in the Leydig cells [12]. •In non-obstructive azoospermia biopsies samples, Mas mRNA, protein, and ACE2 mRNA were lowered compared with biopsies from men with obstructive azoospermia [12].
Ovaries	•Ang II increases the invasive potential of the highly-metastatic ovarian cancer cell line SKOV3 [183]. •RAS components like Ang (1-7) and Mas were localized to primordial, primary, secondary, and antral follicles, stroma, and corpora lutea of reproductive-age ovaries [21]. •High expression of AT1 predicted a shorter survival time for Grade 1 and Grade 2 ovary tumor patients [183].

RAS = Renin-angiotensin system, AT2 = angiotensin II, OVX = ovariectomized rat, TGF-β = transforming growth factor β1, AT4 = angiotensin IV, AT2R = angiotensin II receptor, TNM = Classification of Malignant Tumors, ERK1/2 = extracellular signaling-regulated kinase, MMPs/TIMPs = matrix metalloproteinases/tissue Inhibitor of metalloproteinase, SIF = Sis-inducing factor, AT1 = angiotensin I, OS = overall survival, ACEI/ARB = angiotensin-converting-enzyme inhibitors, ARB = angiotensin receptor blockers, ACE2 = angiotensin-converting enzyme 2.

## Data Availability

Not applicable.

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
