# Peer review of "Angiotensin II and Angiotensin Receptors 1 and 2—Multifunctional System in Cells Biology, What Do We Know?"

_cells, 2021, doi:10.3390/cells10020381_

Round 1

Reviewer 1 Report

In this Review, authors summarized the existing literature knowledge on RAS system, with a focus on cancer association.

Though, numerous articles available in this domain, authors intended to provide an update in this field.

However, the review is interesting, the review needs huge modification before the acceptance for the publication.

  1. Authors should consider summarizing on their own words at the end of each section, that would allow the audience to take a key message.
  2. Many of the points are incomplete and lack of focus. for example:  We observed significant increase in androgen receptor expression in LNCaP cells but 302an insignificant decrease in PC3. -- No reference, and not clear what authors would like to convey.
  3. Consider providing a couple of graphical figures, which provides the info on RAS syatem/mechanism and it's involvement in cancer process.
  4. May consider adding a section on available RAS inhibitors strategy for clinical implications.

Author Response

  1. Authors should consider summarizing on their own words at the end of each section, that would allow the audience to take a key message. Thank you, the suggestion were implemented.
  2. Many of the points are incomplete and lack of focus. for example:  We observed significant increase in androgen receptor expression in LNCaP cells but 302an insignificant decrease in PC3. -- No reference, and not clear what authors would like to convey. Thank you, the appropriate reference was added.
  3. Consider providing a couple of graphical figures, which provides the info on RAS syatem/mechanism and it's involvement in cancer process. Thank you, the figure and new graphic abstract was added.
  4. May consider adding a section on available RAS inhibitors strategy for clinical implications. Thank you, but in our opinion amount of data available on the topic of potential use in therapies is too large. Therefor the potential application of RAS inhibitors may constitute a separate work on this topic.

Reviewer 2 Report

Thank you for the opportunity to read this article.

An important topic which, as clearly shown, influences physiological and pathophysiological processes in many organs.

I think it is a very extensive work with an introduction to the subject and a detailed work-up in the areas beyond the classic understanding of the RAS system.

However, a topic that I consider to be very interesting was only touched upon.

The role of the renin-angiotensin system in liver cirrhosis, in particular portal hypertension and fibrogenesis, is ignored. This would also be an important and interesting aspect in line with the development of hepatocellular carcinoma. For example, it was shown that in experimental cirrhosis, inhibition of chymase leads to natriuretic and hepatic antifibrotic effects, without changes in systemic haemodynamics. This would be an area worth mentioning, for example.

Author Response

Thank you, the suggestion were implemented.

The current version of the chapter 6.3

RAS also plays a significant role in the pathophysiology of the liver. The neoplastic process that leads to the occurrence of hepatocellular carcinoma (HCC) is a complex process and is preceded by two basic pathophysiological conditions: liver inflammation and cirrhosis. The factors leading to hepatitis are mainly hepatitis C virus (HCV), hepatitis B virus (HBV) or alcohol and others [116]. Each of the above mentioned steps is closely related to angiogenesis. Treatment with thioacetamide (TAA) in rat models induced liver fibrosis [117]. Ang II was reported to induce the expression of tumour growth factor β1 (TGF-β1) [117]. The use of perindopril, an ACEI suppressed tumour nectrosis factor α (TNF-α) expression, as compared with TAA group. Moreover, perindopril administration resulted in a limitation of smooth muscle actin alpha 2 (α‐SMA) expression to the portal areas as well as markedly downregulated the expression of AT1R in liver cells and stellate cells [117,118]. In cholangiocarcinoma [119] and hepatocellular carcinoma [120] inhibition of AT1R by telmisartan also resulted in blockage of proliferation. Additionally, it was noticed that telmisartan induced G0/G1 cell cycle arrest by inhibition of G0 to G1 cell cycle transition. However, the use of ACEIs or ARBs showed that no significant difference was observed in HCC risk within 7 years between the initial exposure and non-exposure groups. (HBV and HCV cohorts) [121]. Moreover, there was observed a decrease in the cell cycle-related proteins [119,120] Taken together it might be assumed that RAS role in the pathogenesis of liver diseases is also present, however not fully revealed.

Round 2

Reviewer 1 Report

Authors significantly carried out the corrections for the queries raised by Reviewers.